# Prevalence and risk factors of hypertension in Mauritius: A cross-sectional study

**Rachel Sunny Inyangetuk**[1], **Miguel San Sebastián**[1], **Jaysing Heecharan**[2], **Bhushan Ori**[3], **Paul Zimmet**[4], **Stefan Söderberg**[5], **Jaakko Tuomilehto**[6,7,8,9], **Sudhirsen Kowlessur**[2‡], **Osvaldo Fonseca-Rodríguez**[1‡]*

1 Department of Epidemiology and Global Health, Umeå University, Umeå, Sweden, 2 Non-communicable Disease, Health Promotion and Research Unit, Ministry of Health and Wellness, Port Louis, Mauritius, 3 Ministry of Health and Wellness, Port Louis, Mauritius, 4 Department of Diabetes, Central Clinical School, Monash University, Melbourne, Australia, 5 Department of Public Health and Clinical Medicine, Umeå University, Umeå, Sweden, 6 Population Health Unit, Finnish Institute for Health and Welfare, Helsinki, Finland, 7 Department of Public Health, University of Helsinki, Helsinki, Finland, 8 Saudi Diabetes Research Group, King Abdulaziz University, Jeddah, Saudi Arabia, 9 Department of International Health, National School of Public Health, Instituto de Salud Carlos III, Madrid, Spain

‡ SK and OF-R share the last author position
* osvaldo.fonseca@umu.se

**Data Availability Statement:** Data cannot be shared publicly because of the sensitive nature. Data are available from General Data Protection Regulation (GDPR) Umeå University (Umeå University's Data Protection Officer Email:

## Abstract

Hypertension, a highly prevalent non-communicable disease is a leading cause of death and disability worldwide. In the Republic of Mauritius, the growing number of elderly people along with a rise in overweight and obese populations indicate a looming increase in hypertension prevalence. Given its profound burden on the population and economy, updated insights into the burden and determinants of hypertension in Mauritius is crucial for developing interventions aimed at prevention, management and identifying of at-risk groups. Therefore, this study aimed to estimate the prevalence of hypertension and investigating associated risk factors for hypertension in Mauritius. A cross-sectional study was conducted using the nationally representative data from the 2021 Mauritius non-communicable diseases survey. The survey included 3,622 participants from a total of 4307 contacted individuals (84.1% response rate) aged 19–84 years from all nine districts of the country. The outcome, hypertension, was defined as blood pressure greater than or equal to 140 mmHg systolic and/or 90 mm Hg diastolic or the use of antihypertensive drugs. A log-binomial regression analysis was performed to examine the association between hypertension and a range of sociodemographic, socioeconomic, and behavioural factors, along with the presence of cardiovascular disease. The overall prevalence of hypertension was 35.1%. Older age, low education, high body mass index, occasional or frequent alcohol consumption, and history of cardiovascular disease were significantly associated with hypertension. There is a need to actively implement focused intervention strategies that target and tackle these risk factors to reduce the burden associated with hypertension in Mauritius.

pulo@umu.se) for researchers who meet the criteria for access to confidential data.

**Funding:** The authors received no specific funding for this work.

**Competing interests:** The authors have declared that no competing interests exist.

## Introduction

With a global prevalence of 33% in 2019, hypertension stands as the most prevalent non-communicable disease (NCD), affecting 1.28 billion people, worldwide [1]. Termed as the "silent killer," it accounts for 19.2% of all deaths and is a leading cause of disability [2, 3]. Its impact extends beyond individual health, also negatively affecting individual productivity, families, economies and impeding progress toward the WHO sustainable development goal (SDG) target 3.4 that aims to reduce premature mortality from NCDs by one third [4].

Although hypertension is a significant public health issue, it can be prevented by addressing risk factors associated with it [5]. These include a range of modifiable and sociodemographic factors, such as diet, obesity, alcohol drinking, physical activity, and sleep disorders [6]. Some risk factors such as age and familial predisposition cannot be modified but they may be used for early detection of hypertension. As a result, it becomes evident that the most logical strategy to combat elevated blood pressure lies in effectively controlling the modifiable risk factors [5].

The Republic of Mauritius experiences a significant burden of NCDs that are accounting for 89% of all deaths, with hypertension playing a significant role [7]. Over the last decade, the prevalence of hypertension in Mauritius has remained stable, ranging from 28.4% in 2015 to nearly 30% in 2019 [8]. This lack of improvement has been attributed to a combination of factors, notably a rise in the proportion of overweight or obese individuals (72%), aging of the population that is ranking the highest among sub-Saharan African countries, and a continuous rapid urbanization [9, 10].

Mauritius has integrated its "Mauritius Vision 2030" with the SDGs and is committed to achieving the SDG target of reducing premature mortality from NCDs by one-third by 2030. This goal is set to be achieved through the implementation of its Health Sector Strategic Plan (HSSP) 2020–2024 [11]. However, achieving this target is dependent on effectively addressing the NCD risk factors including hypertension.

While there have been reports from the Mauritius Ministry of Health and Wellness on the prevalence of hypertension at different time periods [10, 12], updated information on the risk factors of hypertension in the Mauritian context is scarce. Studies conducted in the 1990s pointed out to the role of age, overweight and belonging to the Creole ethnicity as important risk factors for hypertension [13, 14]. Additionally, NCDs consume a significant portion of the health expenditure in Mauritius and hypertension is the primary cause of Mauritians attending health facilities [15, 16].

The stable prevalence of hypertension despite existing health policies underscores a critical gap in our understanding of the current risk factors contributing to hypertension in Mauritius. Given the evolving sociodemographic and lifestyle patterns, it is imperative to update and expand the existing knowledge on the prevalence and determinants of hypertension. This study aimed to addressing this gap by investigating the prevalence of hypertension and associated factors in Mauritius in 2021. By identifying and analyzing these factors, our research will offer crucial insights necessary for tailoring public health strategies and interventions, and thus address health inequalities in the country [17].

## Materials and methods

### Study setting

Mauritius is situated in the southwestern region of the Indian Ocean off the east coast of Africa. It is classified as an upper middle-income country and has a population of 1.3 million

people [9]. It is known for its diverse ethnic population, which consists of majorly Indo-Mauritians (70%), Creoles (27%), and Sino-Mauritians (3%) [9, 18].

## Study design

We performed a cross-sectional study using the nationally representative data from 2021 Mauritius NCDs survey where 3,622 respondents (response rate 84.1%) aged 19–84 years participated. Sampling involved a two-stage cluster method ensuring representation of all nine districts in Mauritius, with the number of participants proportionate to each district's population size. Additionally, the sampling also considered regions' urban/rural ratios to obtain a representative sample from all socioeconomic groups.

All personnel who participated in administering the survey were trained health workers, mainly nurses from the Mauritian Ministry of Health and Wellness or government hospitals. The week before the survey, a workshop was held where every participant was specifically trained, first theoretically and then practically, under the supervision of the principal investigators. Each morning, all devices were tested, and those showing outlying values were not used. Detailed information regarding the questionnaires and sampling methods can be found in the Mauritius NCDs survey 2021 report [10]. The Mauritian Ministry of Health and Wellness, in partnership with international research institutions, has conducted a total of seven population surveys on non-communicable disease risk factors since 1987.

**Dependent variable.** Hypertension was the dependent variable for this study, which was defined as the participants reporting use of blood pressure lowering drugs or having blood pressure readings equal to or exceeding 140 mmHg systolic and/or 90 mm Hg diastolic. Trained personnel followed the standardized protocol to measure participants' blood pressure during the survey. Before measurement, participants rested for five minutes and the Omron M7 automatic blood pressure monitor, fitted with different properly sized cuffs, was used for readings. Three consecutive readings, taken at one-minute intervals, and the two final readings were averaged to determine participant's actual blood pressure used in the present study.

**Independent variables.** This study comprised sociodemographic, socioeconomic, health behaviour risk factors, and co-morbidities as potential determinants of hypertension. Sociodemographic factors included sex, age, marital status, ethnicity and place of residence. Sex was divided into males and females, age was stratified into five groups; 19–44 years, 45–54 years, 55–64 years, 65–74 years, and 75 years and above); marital status, was classified into four groups—single/never married, married/cohabiting, separated/divorced, and widowed, while the ethnic groups were categorized into Indo-Mauritians, Creole and Sino-Mauritians. Residence was categorized into urban and rural areas.

Education, occupation and income were part of the socioeconomic factors. Educational level was divided into three categories—primary or no formal education (never attended school and elementary school), secondary education, and tertiary/diploma; the study included 13 occupational groups, which were reclassified into eight for analytical purposes, including professionals, associate professionals and tradespersons, clerical workers, manual workers, students, unemployed, housewives, and retired individuals. Income levels were determined by participants' self-reported average monthly income in Mauritius rupees (1 Euro = 50 rupees) and organized into five groups: richest (above Rs 50,000), richer (Rs 35,001–50,000), middle (Rs 20,001–35,000), poor (Rs 10,001–20,000), and poorest (up to Rs 10,000).

The behavioural risk factors used in this study included smoking, body mass index (BMI), physical activity and frequency of alcohol intake. Smoking status was assessed with the question, "Do you smoke cigarettes, cigars, or pipes? (1. no, 2. ex-smoker for ≥ 3

months, 3. currently a smoker)". Responses were grouped into smokers (if the respondent answered "currently a smoker") and non-smokers (if the respondent answered "no" or "ex-smoker for $\geq$ 3 months"). The participant's BMI (kg/m$^2$) was calculated based on their height and weight measurements. Their height was measured using a stadiometer without shoes, with a precision of 0.1 cm and weight was measured without shoes and excess clothing, with a precision of 0.1 kg using weighing scales. The cut-offs for BMI followed the World Health Organization (WHO) standards [19] (underweight <18.5, normal 18.50–24.99, overweight $\geq$ 25, obese $\geq$ 30) for the Creole population and the adapted to the Asian population (underweight <18.5, normal 18.50–22.99, overweight $\geq$ 23, obese $\geq$ 27.5) for the rest [20].

This study utilized the Global Physical Activity Questionnaire (GPAQ) to collect data on physical activity, measured in metabolic equivalent (MET) minutes per week. The questionnaire assessed moderate and vigorous activity during work, leisure time, and daily commute (walking or cycling to and from work). According to WHO recommendation, individuals should aim for at least 150 minutes of moderate-intensity physical activity or 75 minutes of vigorous-intensity physical activity each week, or a combination of both, resulting in a minimum of 600 METs per week [21]. Participants in this study were classified as physically active if they achieved a minimum of 600 METs per week, while those who attained less than this threshold were classified as physically inactive.

The frequency of alcohol intake was captured by asking participants about their consumption of beer, wine, or spirits, resulting in six initial groups: non-drinkers, ex-drinkers (> 6 months), occasional drinkers, once-weekly drinkers, 2–3 days per week drinkers, and 4 or more days per week drinkers. These groups were combined into three categories: non-drinkers (non-drinkers and ex-drinkers); occasional drinkers (occasional and once-weekly drinkers); and frequent drinkers (participants who drank alcohol 2–3 days and 4 or more days per week).

For co-morbidities, the study participants were categorized into two groups; those who reported to have had a cardiovascular disease (including stroke, angina, myocardial infarction, or coronary artery bypass graft) prior to the survey and those who did not.

## Statistical analysis

Participant characteristics were subjected to descriptive analysis, presented in frequencies and percentages, and stratified based on the outcome variable. The analysis utilized data from 3,374 individuals after excluding respondents 15 who had missing hypertension data and 233 how had missing data from other variables in the survey. Log-binomial regression analyses were used to determine the risk factors associated with hypertension, calculating both unadjusted and adjusted prevalence ratios, and their corresponding 95% confidence intervals for inferential purposes. The data were analysed using R version 4.2.3.

## Ethical considerations

The 2021 Mauritius NCD survey was conducted by the Ministry of Health and Wellness between October 23$^{rd}$, 2021 and November 23$^{rd}$, 2021, receiving ethical approval from the National Ethical Committee (MHC/CT/NETH/2020). Additionally, written informed consent was obtained from all participants in the survey. The management of the data in Sweden was approved by the Swedish Ethical Review Authority (2023-07199-01). The study complied with the Helsinki declaration and reported in line with the Strengthening the Reporting of Observational Studies in Epidemiology (STROBE) guideline.

## Results

### Participants' characteristics

The study included 3,374 participants, 53.6% females and 46.4% males. Most of the respondents were aged 19–44 years (36.9%), married (76.0%), of Indo-Mauritian ethnicity (79.7%) and resided in rural areas (63.0%). Regarding education, more than half (54.0%) had completed secondary education. White-collar workers (including professionals, associate professionals and clerical workers) constituted 31.2% of the sample, while manual workers made up 25.0%. Of the respondents 55.0% were poor or belong to the poorest group and only a minority (7.2%) belonged in the richest group (Table 1).

An analysis of the behavioural factors showed that a moderate proportion of the respondents were smokers (15.5%) and 71.9% of the participants were overweight or obese. Furthermore, 64.8% were reported to be physically active, while 59.0% did not consume alcohol. Only 3.9% of the sample reported having history of cardiovascular diseases (Table 1).

### Prevalence of and risk factors for hypertension

Table 2 presents the prevalence of hypertension according to the different study variables. The prevalence of hypertension in the overall sample was 35.2%, slightly more common among females (35.5%) than in males (34.7%), and consistently increasing with age. Widowed, the Creole, those with primary or no formal education only, housewives and retired, the poorest, those obese, frequent drinkers and with a previous CVD presented a prevalence of hypertension higher than 40%.

Table 2 also shows the results of the regression analyses, displaying both crude and adjusted prevalence ratios. In the crude model, all independent variables except sex and place of residence were associated with hypertension. After adjustment for the significant variables in the crude model, age, ethnicity, education, BMI, alcohol intake and history of CVD remained statistically significantly associated with hypertension. After the age of 45 years, a statistically significant graded increase in the risk of hypertension, from 2.53 among 45–54 year-old to 4.71 in the oldest group (75+) was observed. Also, Creoles had a higher prevalence of hypertension compared with the other ethnic groups (PR = 1.20; 95% CI: 1.08–1.34). Physical activity was inversely associated with the risk of hypertension in the crude analysis but no longer after adjustments.

Individuals with secondary and primary or no formal education had a 42% (PR = 1.42, 95% CI: 1.09–1.84) and 56% (PR = 1.56, 95% CI: 1.19–2.04) higher prevalence, respectively, compared with those with tertiary education. In addition, overweight and obese individuals had a prevalence 34% (PR = 1.34, 95% CI: 1.16–1.54) and 57% (PR = 1.57, 95% CI: 1.34–1.83) higher compared with normal-weight individuals. Similarly, occasional or frequent alcohol drinkers had a statistically significant higher prevalence of hypertension (PR = 1.18; 95% CI: 1.08–1.30 and PR = 1.47; 95% CI: 1.25–1.73, respectively) than non-drinkers. Hypertension was also more prevalent (PR = 1.39; 95% CI: 1.11–1.71) among individuals with history of CVD than those without. Interestingly, people who smoked daily had a lower prevalence of hypertension compared with non-smokers in adjusted analyses (PR = 0.85; 95% CI: 0.73–0.99).

## Discussion

Our study found that the prevalence of hypertension was 35.1% and that older groups, the Creole ethnicity, those with low education, high frequent alcohol intake as well as the presence of CVD were the factors most strongly associated with hypertension. While the observed prevalence is comparable to that in surrounding countries, such as Comoros Island (33.1%) and

**Table 1. Participants´ characteristics, Mauritius 2021.**

| Variable | Frequency (N = 3,374) | Percentage (%) |
|---|---|---|
| **Sex** | | |
| Females | 1,810 | 53.6 |
| Males | 1,564 | 46.4 |
| **Age** | | |
| 19–44 | 1,246 | 36.9 |
| 45–54 | 750 | 22.2 |
| 55–64 | 855 | 25.3 |
| 65–74 | 494 | 14.6 |
| 75+ | 29 | 0.9 |
| **Marital status** | | |
| Single | 577 | 17.1 |
| Married | 2,565 | 76.0 |
| Separated/Divorced | 78 | 2.3 |
| Widowed | 154 | 4.6 |
| **Ethnicity** | | |
| Indo-Mauritian | 2,690 | 79.7 |
| Creole | 501 | 14.8 |
| Sino-Mauritian | 183 | 5.4 |
| **Residence** | | |
| Rural | 2,126 | 63.0 |
| Urban | 1,248 | 37.0 |
| **Education** | | |
| Tertiary | 438 | 13.0 |
| Secondary | 1,821 | 54.0 |
| Primary or no formal education | 1,115 | 33.0 |
| **Occupation** | | |
| Professionals | 344 | 10.2 |
| Assoc Professionals and Trade persons | 460 | 13.6 |
| Clerical workers | 249 | 7.4 |
| Manual workers | 843 | 25.0 |
| Students | 50 | 1.5 |
| Housewife | 563 | 16.7 |
| Unemployed | 131 | 3.9 |
| Retired | 734 | 21.8 |
| **Income** | | |
| Richest | 244 | 7.2 |
| Richer | 339 | 10.0 |
| Middle | 932 | 27.6 |
| Poor | 1,331 | 39.4 |
| Poorest | 528 | 15.6 |
| **Smoking** | | |
| No | 2,851 | 84.5 |
| Yes | 523 | 15.5 |
| **Body mass index** | | |
| Underweight | 163 | 4.8 |
| Normal | 785 | 23.3 |
| Overweight | 1,249 | 37.0 |

(*Continued*)

**Table 1.** (Continued)

| Variable | Frequency (N = 3,374) | Percentage (%) |
|---|---|---|
| Obese | 1,177 | 34.9 |
| **Physical activity** | | |
| No | 1,188 | 35.2 |
| Yes | 2,186 | 64.8 |
| **Frequency of alcohol intake** | | |
| Non-drinkers | 1,991 | 59.0 |
| Occasional drinkers | 1,188 | 35.2 |
| Frequent drinkers | 195 | 5.8 |
| **CVD** | | |
| No | 3,243 | 96.1 |
| Yes | 131 | 3.9 |

Madagascar (36.8%), it is lower than other upper-middle-income countries in sub-Saharan Africa such as South Africa (44%), Namibia (43.7%), and Botswana (43.6%) [1].

Hypertension can vary among African countries due to differences in genetic factors, dietary habits, healthcare access, socioeconomic status, and lifestyle choices. A recent study assessing health systems challenges to control NCDs in Mauritius revealed a suboptimal coverage of population-based interventions combatting main NCD risk factors (tobacco smoking, harmful alcohol use, unhealthy diet and physical inactivity) [22], which could partly explain the lack of decrease of the hypertension prevalence over time.

However, the prevalence of hypertension observed in this study was higher than the prevalence of 28.4% reported in the 2015 Mauritius NCD National survey [12]. This raises concerns and calls for targeted interventions and public health measures to address this growing burden [1]. Hypertension in our study increased steadily as people got older as expected from the literature [23–26] As individuals grow older, the stiffness of the aorta and arterial walls tends to increase contributing to the rise of blood pressure [27].

The higher prevalence of hypertension in the Creole ethnic group compared to Indo-Mauritians has also been reported previously [13]. Nonetheless, the underlying causes of this observation remain unclear, potentially involving genetic predispositions and varying dietary practices. This study also demonstrated a strong association between low education and hypertension. This finding is consistent with a study from Korea [28] and a meta-analysis from South Asian countries [6]. Lower education levels may lead to poor health literacy [29]. It has also been shown that education is the most reliable socioeconomic indicator associated with a risk of major CVD events [30].

The association with overweight/obesity was expected since previous studies also from Mauritius had found this association [31]. This has been commonly observed in many international studies in this field; for instance, a meta-analysis involving 43,025 adults from 15 African countries found overweight/obesity to be independently associated with hypertension [32]. Excess body weight may lead to hypertension through impaired structural and functional renal activity, increased cardiac output, arterial and aortic stiffening due to the activation of sympathetic activity and the renin-angiotensin-aldosterone system (RAAS) [33]. Research indicates that excess body weight plays a significant role in the development of hypertension, accounting for 65–75% of the risk for primary hypertension [6].

Likewise, it has been shown that regular consumption of moderate to high levels of alcohol is associated with an elevated risk of hypertension [34–36], but consuming low amounts of

**Table 2. Prevalence of hypertension and crude and adjusted\* prevalence ratios of the association between the social risk factors and hypertension (95% confidence intervals in brackets), Mauritius 2021.**

| Variable | Prevalence (N = 1,186) | Crude Prevalence Ratio (95%CI)[1] | Adjusted\* Prevalence Ratio (95%CI)[1] |
|---|---|---|---|
| **Sex** | | | |
| Females | 643 (35.5%) | — | |
| Males | 543 (34.7%) | 0.98 (0.89, 1.07) | |
| **Age** | | | |
| 19–44 | 151 (12.1%) | — | — |
| 45–54 | 271 (36.1%) | 2.98 (2.50, 3.56) | 2.53 (2.11, 3.04) |
| 55–64 | 441 (51.6%) | 4.26 (3.62, 5.01) | 3.64 (3.06, 4.33) |
| 65–74 | 302 (61.1%) | 5.04 (4.28, 5.95) | 4.30 (3.55, 5.22) |
| 75+ | 21 (72.4%) | 5.98 (4.56, 7.83) | 4.71 (3.57, 6.22) |
| **Marital status** | | | |
| Single | 101 (17.5%) | — | — |
| Married | 969 (37.8%) | 2.16 (1.80, 2.59) | 1.09 (0.92, 1.29) |
| Separated/Divorced | 31 (39.7%) | 2.27 (1.64, 3.15) | 1.24 (0.91, 1.68) |
| Widowed | 85 (55.2%) | 3.15 (2.51, 3.96) | 1.04 (0.84, 1.29) |
| **Ethnicity** | | | |
| Indo-Mauritian | 910 (33.8%) | — | — |
| Creole | 206 (41.1%) | 1.22 (1.08, 1.37) | 1.20 (1.08, 1.34) |
| Sino-Mauritian | 70 (38.3%) | 1.13 (0.93, 1.37) | 0.96 (0.79, 1.16) |
| **Residence** | | | |
| Rural | 722 (34.0%) | — | |
| Urban | 464 (37.2%) | 1.09 (1.00, 1.20) | |
| **Education** | | | |
| Tertiary | 56 (12.8%) | — | — |
| Secondary | 586 (32.2%) | 2.52 (1.95, 3.24) | 1.42 (1.09, 1.84) |
| Primary or no formal education | 544 (48.8%) | 3.82 (2.97, 4.91) | 1.56 (1.19, 2.04) |
| **Occupation** | | | |
| Professionals | 53 (15.4%) | — | — |
| Associate Professionals and Trade persons | 126 (27.4%) | 1.78 (1.33, 2.37) | 1.20 (0.90, 1.59) |
| Clerical workers | 57 (22.9%) | 1.49 (1.06, 2.08) | 0.99 (0.72, 1.37) |
| Blue collar workers | 297 (35.2%) | 2.29 (1.76, 2.98) | 1.19 (0.91, 1.55) |
| Students | 6 (12.0%) | 0.78 (0.35, 1.72) | 0.97 (0.50, 1.85) |
| Housewife | 231 (41.0%) | 2.66 (2.04, 3.48) | 1.20 (0.91, 1.58) |
| Unemployed | 46 (35.1%) | 2.28 (1.62, 3.20) | 1.19 (0.85, 1.65) |
| Retired | 370 (50.4%) | 3.27 (2.53, 4.24) | 1.20 (0.92, 1.57) |
| **Income** | | | |
| Richest | 56 (23.0%) | — | — |
| Richer | 78 (23.0%) | 1.00 (0.74, 1.36) | 0.97 (0.73, 1.28) |
| Middle | 311 (33.4%) | 1.45 (1.14, 1.86) | 1.13 (0.89, 1.43) |
| Poor | 494 (37.1%) | 1.62 (1.27, 2.06) | 1.07 (0.84, 1.36) |
| Poorest | 247 (46.8%) | 2.04 (1.59, 2.61) | 1.19 (0.93, 1.53) |
| **Smoking** | | | |
| No | 1,056 (37.0%) | — | — |
| Yes | 130 (24.9%) | 0.67 (0.57, 0.78) | 0.85 (0.73, 0.99) |
| **Body mass index** | | | |
| Underweight | 21 (12.9%) | 0.49 (0.33, 0.75) | 0.54 (0.37, 0.79) |
| Normal | 205 (26.1%) | — | — |

*(Continued)*

**Table 2.** (Continued)

| Variable | Prevalence (N = 1,186) | Crude Prevalence Ratio (95%CI)[1] | Adjusted* Prevalence Ratio (95%CI)[1] |
|---|---|---|---|
| Overweight | 444 (35.5%) | 1.36 (1.18, 1.56) | 1.33 (1.17, 1.51) |
| Obese | 516 (43.8%) | 1.68 (1.47, 1.92) | 1.78 (1.57, 2.02) |
| **Physical activity** | | | |
| No | 453 (38.1%) | — | — |
| Yes | 733 (33.5%) | 0.88 (0.80, 0.97) | 1.00 (0.91, 1.09) |
| **Frequency of alcohol intake** | | | |
| Non-drinkers | 689 (34.6%) | — | — |
| Occasional drinkers | 410 (34.5%) | 1.00 (0.90, 1.10) | 1.18 (1.08, 1.30) |
| Frequent drinkers | 87 (44.6%) | 1.29 (1.09, 1.52) | 1.47 (1.25, 1.73) |
| **CVD** | | | |
| No | 1,095 (33.8%) | — | — |
| Yes | 91 (69.5%) | 2.06 (1.82, 2.33) | 1.39 (1.23, 1.58) |

*Adjustment done for statistically significant variables in the crude model.

alcohol or none is not necessarily linked to a reduced risk of hypertension [5]. Similarly, longitudinal studies carried out in South Korea [36] and Japan [37] have reported results that are in line with our findings, indicating a direct association between high alcohol consumption and hypertension risk among both males and females. Alcohol has been shown to increase the risk of hypertension by increasing sympathetic nervous system activity, leading to increased heart rate and vasoconstriction, it also diminishes baroreceptor sensitivity, which impairs blood pressure regulation. Additionally, alcohol stimulates the RAAS, promoting sodium and water retention, leading to increased blood volume and peripheral resistance [38, 39].

Considering hypertension is a primary risk factor for CVD [37], it was expected to find a high prevalence of hypertension among individuals with history of CVD, as it is probable that hypertension occurred before the development of the disease.

An intriguing finding from our analysis was the absence of association between hypertension and classical risk factors such as smoking and physical activity. The lack of significance could be potentially explained by the moderator effect of age, as smoking was more common among the youths, which had the lowest prevalence of hypertension in our sample. Also, smokers have been shown to have lower blood pressure possibly due to renal hyperfiltration as seen in a Swedish cohort study [40]. This observation aligns with similar studies conducted in Uganda and Nepal [41, 42]. Likewise, age moderated the association between physical activity and hypertension, as physical activity was more common among the youngest group who presented the lowest prevalence of hypertension. Similar results have been seen in a study conducted by Cheah et al. [43] in Malaysia.

## Strengths and limitations of the study

This study utilized the recent 2021 Mauritius NCD Survey which is based on a nationally representative dataset that covers both sexes, all regions, and ethnic groups in the country. The study employed a standardized and comprehensive methodology to capture exposure and outcome variables including the careful and standardized measurement of blood pressure. As a result, our findings reflect the current situation in sociodemographic patterns and lifestyle practices in Mauritius that can be widely applicable to the broader population.

However, the study has some limitations that need to be considered. This study could not directly test the effects of other potentially influencing factors of hypertension such as stress levels, dietary habits or birth weight. Since all lifestyle factors were self-reported, recall bias, but also social desirability, could have been present, leading to an over or underestimation of the reported factors. Also, given the cross-sectional design of the study, the observed associations should not be interpreted as causal.

## Conclusions

The overall prevalence of hypertension in Mauritius was 35.1%, revealing older age, the Creole ethnic group, lower education level, overweight and obesity, elevated alcohol consumption, and the presence of CVD as significant factors associated with hypertension in Mauritius. This study highlights the need for targeted interventions to address the growing prevalence of hypertension in Mauritius. Concerted efforts by the Mauritian Government should be made to promote health literacy and sustainable strategies that address modifiable risk factors for hypertension across all ages. There is also need for multisectoral collaboration to promote to risk factor awareness and emphasize primary and secondary prevention, especially among at-risk groups such as the elderly and Creole ethnicity. Additionally, further research is needed into the socioeconomic factors associated with hypertension and the barriers to the adherence with healthcare use for screening and treatment in this country with cost-free health services.

## Acknowledgments

We wish to thank all participants in this study for taking their time to complete the questionnaire.

## Author Contributions

**Conceptualization:** Stefan Söderberg.

**Data curation:** Jaysing Heecharan, Bhushan Ori, Sudhirsen Kowlessur.

**Formal analysis:** Rachel Sunny Inyangetuk, Osvaldo Fonseca-Rodríguez.

**Methodology:** Miguel San Sebastián.

**Writing – original draft:** Rachel Sunny Inyangetuk, Osvaldo Fonseca-Rodríguez.

**Writing – review & editing:** Miguel San Sebastián, Jaysing Heecharan, Bhushan Ori, Paul Zimmet, Stefan Söderberg, Jaakko Tuomilehto, Sudhirsen Kowlessur.

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
