## [Decision Letter · Decision Letter 0]

24 May 2024

PGPH-D-24-00818

Prevalence and risk factors of hypertension in Mauritius: a cross-sectional study

Dear Dr. Fonseca Rodríguez,

Thank you for submitting your manuscript to PLOS Global Public Health. After careful consideration, we feel that it has merit but does not fully meet PLOS Global Public Health’s publication criteria as it currently stands. Therefore, we invite you to submit a revised version of the manuscript that addresses the points raised during the review process.

We look forward to receiving your revised manuscript.

Kind regards,

Tarra L Penney

Academic Editor

Journal Requirements:

Additional Editor Comments (if provided):

Reviewers' comments:

Reviewer's Responses to Questions

**Comments to the Author**

1. Does this manuscript meet PLOS Global Public Health’s publication criteria? Is the manuscript technically sound, and do the data support the conclusions? The manuscript must describe methodologically and ethically rigorous research with conclusions that are appropriately drawn based on the data presented.

Reviewer #1: Yes

Reviewer #2: Yes

Reviewer #3: Yes

2. Has the statistical analysis been performed appropriately and rigorously?

Reviewer #1: Yes

Reviewer #2: I don't know

Reviewer #3: I don't know

3. Have the authors made all data underlying the findings in their manuscript fully available (please refer to the Data Availability Statement at the start of the manuscript PDF file)?

Reviewer #1: Yes

Reviewer #2: No

Reviewer #3: Yes

4. Is the manuscript presented in an intelligible fashion and written in standard English?

Reviewer #1: Yes

Reviewer #2: Yes

Reviewer #3: Yes

5. Review Comments to the Author

Reviewer #1: Review comments – PLOS Global Public Health

The paper is interesting and could contribute to bridging literature gap on hypertension prevalence and risk factors in Mauritius. However, the manuscript does not merit publication at it’s current state.

Abstract

Lines 42-43: A cross-sectional national representative data obtained from the 2021 Mauritius non-communicable diseases survey was used for this study. “This statement does not improve readability. I suggest the authors revise the statement to “A cross-sectional study was conducted using nationally representative data from the 2021 Mauritius non-communicable diseases survey”.

Introduction

Lines 73 – 78: The lines 73 – 78 talks about the prevalence and risk factors associated with hypertension in Mauritius. If you are aware of the risk factors of hypertension in Mauritius, what is the motivation for conducting this study?

Lines 88 – 94: Lines 88 – 94 emphasise on the justification of the study. The authors should strengthen their study justification as it is weak in its current state. Why are they conducting the study? Is it to bridge literature gap in Mauritius?

Materials and Methods

There is no sub-section called “study design” in the methods section which is major weakness of the study. The authors should consider having a separate paragraph to present on study design.

Study setting

The information on study setting is too scanty. The authors can add on.

Dependent variables

Lines 111 – 118: Trained personnel followed the standardized protocol to measure participants' blood pressure during the survey. Before measurement, participants rested for five minutes and the Omron M7 automatic blood pressure, monitor, fitted with different properly sized cuffs, was used for readings. Three consecutive readings, taken at one-minute intervals, and the two closest values were averaged to determine participant’s actual blood pressure used in the present study.

The authors reported that they relied on secondary data for this study. How did they train people to measure participant’s blood pressure? It is quite confusing

Independent variables

Line 123: categorization of age was reported as 19 -<45 years, 45– <55 years, 55 -<65 years, 65-<75 years. Why not write as 19 – 45, 45 – 55 and 55 – 65 years. The question is, if someone is exactly 45 years which category will be person be placed 19-45 category or 45 – 55 category? This applies to the remaining categorizations. It needs to be done well.

Additionally, is there any justification for wide variations in the categories comparing the first category (19-45 yrs) to the others.

Line 128: It is not allowed to combine “no formal education and elementary” as primary education. One who had “no formal education should be treated as had “no formal education”.

Statistical analysis

The abstract is indicating 3622 as participants and the statistical analysis section is indicating 3607. Could you speak to the differences in the sample?

Ethical considerations

The authors should indicate clearly if they were involved in the data collection exercise of the NCD survey. If yes, they should write on the detailed process of the informed consent process in addition to the ethics. If not, they should state that they were not involved in the data collection exercise and only relied on secondary data from the NCD survey. Then, they can state where (IRB) the NCD survey had ethical approval from and whether the study adhered to the declaration of Helsinki.

Results

Line 190: There is no very poor in Table 1 so I find it difficult to get where the very poor is from.

Lines 234 – 245: Can the authors justify what could contribute to the differences in the hypertension prevalence compared with other countries.

Line 259: How does excess body weight contributes to hypertension. Kindly add to the discussion.

Line 265: Just indicating that literature supports that alcohol consumption is linked to hypertension is not enough. Kindly add how it contributes to hypertension

Lines 270: It appears misleading to commence the statement with an intriguing finding. This posits that you had a preconceived results you wanted to see or report on.

Conclusion

Lines 295 – 296 have no relation with the conclusion, kindly scrap it. I suggest the authors strengthen their concluding statement and prevent repetition of results.

Implications for practice and research is completely missing.

Reviewer #2: This paper reports data from a health survey in Mauritius. The data that are reported include prevalence of hypertension and factors associated with hypertension, limited by the survey questions and respondent self-reporting.

This report provides valuable insights into hypertension in Mauritius and further information on data from Sub-Saharan Africa.

Comments:

1. In general, the paper is well written. Minor language editing is required: the study aimed "to" (rather than "at"), "A cross-sectional national representative data" - presumably database?,

2. The first sentence in the abstract may not be needed.

3. Study setting might be better placed in the introduction.

4. You comment that lower education is associated with hypertension. However, the prevalence is lower in the lower education group (12.8% vs 32% and 48%) and the prevalence ratio is higher with higher education. Please clarify and discuss this further.

Reviewer #3: I would like to suggest that the authors mention the approval number provided by Ethical Board Review Board of the country to conduct the survey. This study is not showing extraordinary report and delivers what is already known to the world. Hence, the co-relation of the stratified age group with smoking cigarettes, physical activity and hypertension would add quality to the paper.

6. PLOS authors have the option to publish the peer review history of their article (what does this mean?). If published, this will include your full peer review and any attached files.

**Do you want your identity to be public for this peer review?** For information about this choice, including consent withdrawal, please see our Privacy Policy.

Reviewer #1: No

Reviewer #2: No

Reviewer #3: **Yes: **Sweta Koirala

---

## [Decision Letter · Decision Letter 1]

30 Sep 2024

PGPH-D-24-00818R1

Prevalence and risk factors of hypertension in Mauritius: a cross-sectional study

Dear Dr. Fonseca Rodríguez,

Thank you for submitting your manuscript to PLOS Global Public Health. After careful consideration, we feel that it has merit but does not fully meet PLOS Global Public Health’s publication criteria as it currently stands. Therefore, we invite you to submit a revised version of the manuscript that addresses the points raised during the review process.

Thank you for submitting your revised manuscript, after careful review additional clarifications are needed before a decision on publication.  Please respond to the remaining points below from reviewer #1.

We look forward to receiving your revised manuscript.

Kind regards,

Tarra L Penney

Academic Editor

Journal Requirements:

Additional Editor Comments (if provided):

Reviewers' comments:

Reviewer's Responses to Questions

**Comments to the Author**

1. If the authors have adequately addressed your comments raised in a previous round of review and you feel that this manuscript is now acceptable for publication, you may indicate that here to bypass the “Comments to the Author” section, enter your conflict of interest statement in the “Confidential to Editor” section, and submit your "Accept" recommendation.

Reviewer #1: No

Reviewer #2: All comments have been addressed

Reviewer #3: All comments have been addressed

2. Does this manuscript meet PLOS Global Public Health’s publication criteria? Is the manuscript technically sound, and do the data support the conclusions? The manuscript must describe methodologically and ethically rigorous research with conclusions that are appropriately drawn based on the data presented.

Reviewer #1: Yes

Reviewer #2: Yes

Reviewer #3: Partly

3. Has the statistical analysis been performed appropriately and rigorously?

Reviewer #1: Yes

Reviewer #2: I don't know

Reviewer #3: I don't know

4. Have the authors made all data underlying the findings in their manuscript fully available (please refer to the Data Availability Statement at the start of the manuscript PDF file)?

Reviewer #1: Yes

Reviewer #2: No

Reviewer #3: Yes

5. Is the manuscript presented in an intelligible fashion and written in standard English?

Reviewer #1: Yes

Reviewer #2: Yes

Reviewer #3: Yes

6. Review Comments to the Author

Reviewer #1: Thanks for the opportunity to re-review the manuscript. There is still a confusing in the manuscript; at some points authors report they use secondary data, at some points they report on data collection as if they actually collected primary data. The authors should clarify this by reporting as a study that utilised solely secondary data.

Abstract

The survey included 3,622 participants aged 19–84 years from all nine districts of the country. Please, report the sample size you used for the analysis in the abstract.

Materials and methods

Study settings

Could you add some information to the study setting?

Study design

We performed a cross-sectional study using the nationally representative data from 2021 100 Mauritius NCDs survey where 3,622 respondents (response rate 84.1%). I think your confusing stems from the fact that some of the authors participated in the survey. Once you stated that you used secondary data, it remains secondary data. As such, report on the data you used for the analysis and state why you excluded some of the participants where necessary.

Independent variables

Sex was divided into men and women. Revise to “male and female”

Results

Table 1

In the previous comment, I talked about how the table 1 should be presented, we keep going back. It is appropriate to have the % on the header section so you do not repeat in the entire table.

For example.

Variable Frequency (n=3,607) Percentage (%).

I do not know how you computed the prevalence. But, I had a different prevalence. If 1186 participants had hypertension out of 3,607, the overall prevalence cannot be 35.2%, it should be 32.9%.

Lines 230 – 239, please check and write the interpretation of the prevalence ratio well. I do not think you need to add % to the prevalence ratio when interpreting.

Study limitations

The study employed a standardized and comprehensive methodology to capture exposure and outcome variables including the careful and standardized measurement of blood pressure. Did you collect the data? You have reported that the study relied on secondary data, kindly report as such.

Reviewer #2: Thank you for addressing my comments.

Reviewer #3: n/a

7. PLOS authors have the option to publish the peer review history of their article (what does this mean?). If published, this will include your full peer review and any attached files.

**Do you want your identity to be public for this peer review?** For information about this choice, including consent withdrawal, please see our Privacy Policy.

Reviewer #1: No

Reviewer #2: No

Reviewer #3: No

---

## [Editor Report · Decision Letter 2]

14 Nov 2024

Prevalence and risk factors of hypertension in Mauritius: a cross-sectional study

PGPH-D-24-00818R2

Dear Dr Fonseca Rodríguez,

We are pleased to inform you that your manuscript 'Prevalence and risk factors of hypertension in Mauritius: a cross-sectional study' has been provisionally accepted for publication in PLOS Global Public Health.

Best regards,

Tarra L Penney

Academic Editor

Reviewer Comments (if any, and for reference):

No additional comments, revision were accepted.